# Disentangling the effects of modality, interval length and task difficulty on the accuracy and precision of older adults in a rhythmic reproduction task

**Elisa M. Gallego Hiroyasu**[ID]**, Yuko Yotsumoto**[ID]*

Department of Life Sciences, The University of Tokyo, Tokyo, Japan

* yuko@fechner.c.u-tokyo.ac.jp

**Data Availability Statement:** All data and model comparison result files are available from the OSF database (DOI 10.17605/OSF.IO/2N9S8).

## Abstract

Studies on the functional quality of the internal clock that governs the temporal processing of older adults have demonstrated mixed results as to whether they perceive and produce time slower, faster, or equally well as younger adults. These mixed results are due to a multitude of methodologies applied to study temporal processing: many tasks demand different levels of cognitive ability. To investigate the temporal accuracy and precision of older adults, in Experiment 1, we explored the age-related differences in rhythmic continuation task taking into consideration the effects of attentional resources required by the stimulus (auditory vs. visual; length of intervals). In Experiment 2, we added a dual task to explore the effect of attentional resources required by the task. Our findings indicate that (1) even in an inherently automatic rhythmic task, where older and younger adult's general accuracy is comparable, accuracy but not precision is altered by the stimulus properties and (2) an increase in task load can magnify age-related differences in both accuracy and precision.

## Introduction

In our daily lives, we often time ourselves to control and synchronize our movements or predict something that may happen in the future. How our brain measures time is demonstrated with the pacemaker-accumulator model [1]. With this model, the pacemaker of a hypothetical "internal clock" in our brain is considered to beat pulses at a certain rate, and the accumulator stores these pulses. The accumulation of these pulses is what we usually translate into and perceive to be a duration of time.

Previous temporal processing studies have led to different conclusions on the functioning of this pacemaker in older adults. One of the main understandings is that older adults have a slower internal temporal clock than younger adults [2–4]. Nevertheless, other studies suggest that older adults may have faster internal clocks [5–7] or that their clocks do not differ from those of younger adults [8–10]. The cause of the contradicting findings given these mixed results on the temporal cognition of older adults remains unclear.

**Funding:** This research was funded by Japan Society for the Promotion of Sciences (JSPS https://www.jsps.go.jp/english/) KAKENHI (Grant #19H01771, #19H05308, #17K18693) and UTokyo-CiSHuB (https://marler.c.u-tokyo.ac.jp/humanscience/en/) to YY. The funders had no role in study design, data collection and analysis, decision to publish, or preparation of the manuscript.

**Competing interests:** The authors have declared that no competing interests exist.

One possibility for mixed results pertains to the differences in tasks and the cognitive demand of each of them. Until now, age-related differences in timing have been studied in terms of the centralized model [1, 11] where the hypothetical clock assumes the role of the general pacemaker and accumulator for all temporal intervals and sensory modalities. And it is this hypothetical clock that pace at a slower rate in older adults. However, several reasons exist for why the temporal processing of older adults should be studied using the framework of the distributed timing model [11, 12] especially when perceived duration can be manipulated by changing the physical properties of the stimulus [13].

First, it has been suggested that various temporal durations, both under or over one second, are processed by different mechanisms [11, 14, 15]. Further, Piras and Coull [16] found that brief durations of 200ms may recruit more automatic mechanisms that are comparable to those of the implicit task, and longer durations of 600ms and 1400ms would utilize those of the explicit task. Similarly, Bangert and Balota [5] found that longer tap intervals of 1500ms, as opposed to 500ms and 1000ms, engage higher attentional resources. Therefore, intervals over one and a half seconds may magnify differences in the reproduction of an interval due to a bigger demand in attentional resources.

Second, modality differences are also observed in temporal cognition. In accordance with the distributed timing model, auditory and visual signals are measured at different clock speeds [17, 18]. It is well known that auditory modalities are much more temporally precise than visual modalities [19–22] and that pre-exposure to auditory rhythms can facilitate perception in visual modality [23]. Moreover, visual stimuli rely on auditory input [24] even if only to some extent [23]. Visual stimuli are also known to recruit more attentional resources [25], thus enabling bigger differences to be observed between older and younger adults in their visual modalities.

In fact, there are a few studies that compare the temporal processing between different modalities in older adults [20, 26]. Yet, these studies asked older adults to reproduce or judge single intervals. This methodology, however, requires the older adults to hold temporal information in their working memory. It therefore remains unclear whether the alterations in temporal reproductions and judgements is due to inaccurate perception of temporal information or the unsatisfactory preservation of the temporal information in the working memory. Thus, we deemed important to study modality differences, as well as duration differences, to disentangle the possible differences in the perceptual processing of target interval while also considering the use of rhythmic sequences to alleviate the problems with interval memory retention and the attentional resources these may require.

Attentional resources allocated in the perception of time is not only recruited by the different stimulus properties but also in the task instructions. As explained by the attentional gate model [27], these are crucial in temporal cognition. This model sheds light on the importance of attentional resources in temporal perception such that the number of pulses sent to the accumulator depends on how much attentional resources are allocated to the timing task. Later, the encoded durations are transferred into the working memory where this duration can be manipulated.

With age, however, there are numerous cognitive deteriorations [28, 29] including deteriorated attention, memory, and sensory input quality, which can lead to decreased temporal cognition [30]. The downfall of these cognitive factors may affect some of the tasks that are used to investigate temporal processing in older adults and hence, many researchers have attributed the performance differences of older and younger adults to the attention and memory demands of the task [31–33].

Yet, whether all age-related differences can be attributed to the cognitive demands of the task is a difficult question to answer. For example, Block and colleagues [7] found evidence

suggesting faster internal clocks for older adults in production tasks and verbal estimation tasks but their performance in reproduction tasks were similar to those of younger adults. As they explain, "even if the rate of physiological and cognitive processes varies with age, the same rate will subserve a person's experiencing the target duration and reproducing it" (p.586). Yet, Baudouin, Vanneste, Isingrini and Pouthas [34] found that single-interval production and reproduction tasks involve different mechanisms, where only the latter correlates with working memory measures, so age-related differences may arise for reproduction tasks if the temporal information decays in the working memory of older adults.

One way in which we can isolate temporal cognition from other higher-level cognitive factors, apart from those required by the stimulus property itself, is by using rhythmic sequences, rather than single intervals. Beat perception may feel automatic [35] and reproducing the interval of a beat-based rhythm is cognitively less demanding than that of holding temporal information of a single interval in working memory and reproducing it [36]. The predictive nature of beat-based rhythm, as opposed to single intervals, may help older adults compensate for the age-related decline of temporal processing [37]. Though several correlations have been observed between focused attention and beat perception [38], studies have also shown that older adults preserve temporal prediction capabilities when events are temporally fixed [39, 40]. Nonetheless, rhythmic sequences are considered to evoke both sensory and motor representations [41, 42], thus allowing deficits in the production of temporal intervals to appear if structural changes in the brain alters temporal cognition in older adults.

Despite a reduced cognitive load using rhythmic sequences, behavioral studies that involve tapping tasks with older adults have also demonstrated contradictory results. Turgeon and Wing [2] found that the tapping rates of older adults indicate a slower clock even with tasks that eliminate the contribution of higher cognitive functions, besides that of temporal cognition. On the other hand, Bangert and Balota [5] observed faster tapping in older adults, which should in principle indicate a faster clock. Yet, they justified this observation with the participant's difficulty in maintaining the reproduced intervals in memory and difficulties in error-correcting. Thus, the presence of contradicting findings on the speed of the internal clock of older adults raises questions on the performance of older adults and how it compares to that of younger adults.

In this paper, we deemed important to disentangle the effect of cognitive factors that may have repercussions on the true temporal processing ability of older adults: 1) those embedded in the stimulus property, and 2) those required by demanding tasks. First, by using a simple rhythmic task, we aimed to understand how aging may impact temporal cognition while considering possible differences between the physical properties of the stimuli, such as the lengths of temporal intervals and modality differences between auditory and visual stimuli.

To verify age-related differences in pure temporal cognition respective to modalities and longer intervals, in Experiment 1, we asked participants to follow the presented rhythmic pattern to observe whether the reproduced intervals of younger and older adults were comparable. Under the distributed timing model, we predicted that older adults would reproduce durations above 1.5 second and visually marked intervals with less accuracy and precision compared to younger adults, since these stimulus properties recruit different networks [42] and more significant amounts of attentional resources [20, 21]. We further explored whether the age-related difference between stimulus properties would show in the variability of the reproduced intervals, since scalar property is a dominant characteristic supporting the centralized model [1, 11].

Given that in many tasks, recruitment of higher cognitive functions is not only embedded in the stimulus property but also in the task itself, in Experiment 2, we manipulated attentional resources that were recruited in the completion of the rhythmic reproduction task. As

illustrated in the attentional gate model [27], when participants are asked to concurrently do a nontemporal task, attentional resources are divided between the two tasks. This means that less attentional resources are allocated to the timing task and thus, reproduced durations shorten. Since older adults have limited resources compared to younger adults, we further investigated whether a nontemporal secondary task can decrease attentional resources allocated to the timing task and hence provoke bigger age-related differences in both accuracy and precision based on modality and duration.

## Experiment 1

### Methods

**Participants.** An a priori power analysis was conducted using G*Power3.1 [43] to detect within-between interactions given repeated measures. We used 12 as the number of measurements per participant, a large effect size ($f$ = .30), an alpha of .05, and default values for correlations among repeated measures and non-sphericity corrections. A total sample size of 14 was required to achieve a power of .95.

Fifteen older ($M$ = 71.1; $SD$ = 4.02; female = 7, male = 8) adults were recruited from the Third Generation Human Resource Center in Meguro-ward (Tokyo, Japan). To exclude those with signs of cognitive impairments, we ensured that all older adults scored over 25 (*Range* 27~30; $M$ = 29.27) on the Mini-Mental State Examination (MMSE). As a control group, we also recruited 15 younger ($M$ = 21.9; $SD$ = 2.67; female = 6, male = 9) adults from The University of Tokyo. Our participant numbers per age group are similar to Lustig and Meck [20], leading to a total of 30 participants in Experiment 1. Participants from both age groups have self-reported normal hearing and normal or corrected-to-normal vision.

In accordance with the Declaration of Helsinki, all subjects provided written informed consent. The protocol was approved by the institutional review board of The University of Tokyo, and the subjects received monetary awards for their participation.

**Stimuli presentation.** A trial consisted of an isochronous sequence that was presented by five stimuli of either the auditory or visual modality. Auditory beeps (3500Hz with a sampling rate of 6000Hz with a duration of 30ms) were presented at a listening level of 80dB using two speakers (Sanwa Supply: MM-SPL5BK) that were located on both sides of the CRT monitor (SONY-CPD E230) with a refresh rate of 60Hz and a display resolution of 1024 by 768. Visual sequences were marked by five white circles (7.25°diameter) that were presented on a dark grey background. These circles appeared on and off at the center of the screen five times for 16.7 milliseconds each. As for the temporal durations, stimulus-onset asynchrony (SOA) values ranged from milliseconds to seconds, at 300ms, 700ms, 1300ms, 1700ms, 2300ms, and 2700ms.

Participants sat in a dark, soundproof room (1.73m in length by 0.85m in width by 1.92m in height) and were asked to sit comfortably facing the screen at a distance approximately 50cm from their eyes to the CRT monitor. Stimuli presentation and data collection were completed on a Mac Pro (mid-2010) with macOS Sierra using PsychportAudio and PsychToolbox extensions [44] on MATLAB 2017b.

**Procedure.** Participants performed two separate tasks: the auditory (Fig 1A) and the visual task (Fig 1B). In each task, participants were asked to listen or look at five stimuli presented in a rhythmic manner. The trial began with a cue that indicated the initiation of the trial, followed by a black fixation cross on a grey background. The rhythm was marked with five stimuli that were presented isochronously. Though participants were permitted to count the number of times the stimuli had appeared, they were explicitly instructed not to count the time elapsed between each stimulus. When the time came, participants replicated the temporal

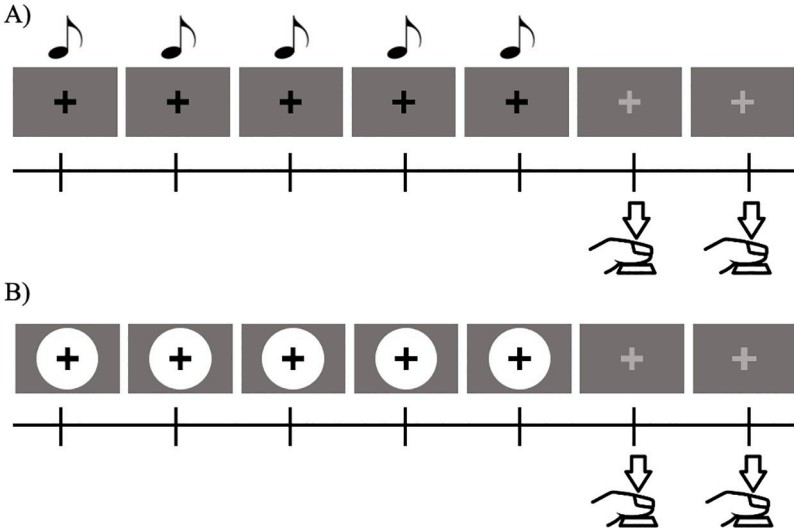

**Fig 1. Experiment 1—Procedure of the reproduction task.** Participants were required to press the button twice while simultaneously monitoring the rhythm of the stimuli. The stimulus-onset asynchrony (SOA) for each trial was either 300ms, 700ms, 1300ms, 1700ms, 2300ms, or 2700ms. The task was performed in the auditory modality (A) and the visual modality (B). Note that the white disks flashed on and off rhythmically five times. In addition, the fixation cross maintained statically on the screen for both modalities.

interval by pressing the "Enter" key of the number keyboard twice. In the case that a participant miscounted the five stimuli, a visual aid (red fixation cross) indicated that the five stimuli had already appeared, and they had to proceed to continuing the rhythm. The red fixation remained statically on the screen during the button press, until the end of the trial.

To avoiding central tendency effect and not to surprise the participant when a short duration followed a long duration, the task for each of the modalities were separated into six blocks, each containing either short (under one sec: 300ms, 700ms), middle (between one and two secs: 1300ms, 1700ms), or long durations (over two secs: 2300ms, 2700ms). We opted this method to be adequate also because temporal preparation has been known to be influenced by the previous stimuli durations [45]. Note that for the rest of the paper, short and long durations do not refer to these categories, rather the continuously increasing length of the intervals.

A total of 60 sequences of each modality were reproduced by participants, ten for each timing condition. Stimuli within blocks and the blocks themselves were randomized for each participant. All participants completed this task for both the auditory and visual modalities. The order in which these tasks were performed was counterbalanced.

**Statistical analysis.** To examine whether differences existed between the two age groups, we analyzed both the accuracy and the precision of the reproduced intervals using Bayesian Statistics on JASP [46]. Unlike the frequentist method, in which a no-significance does not indicate no-difference, Bayesian statistics quantify the evidence for the null hypothesis [47]. Since the Bayes Factors may not seem to be a clear-cut statistical methodology like the classical frequentist method, we report both Bayes Factors ($BF_{10}$) and the frequentist result. Yet, we interpret our data based on $BF_{10}$ which compare all alternative models against the null model. Bayesian statistics is also competent for it is less dependent on the sampling intentions [48] and more capable of converging to the correct decision compared to the frequentist method [49]. A $BF_{10}$ equal to value X means that the alternative hypothesis is X times more likely than that of the null hypothesis.

It is also important to note that only the result of the best performing model is shown on this paper for there are too many models from the model comparison to list. Yet, as they are considered important for the interpretations of the results, all data, analysis scripts, and results, including the results from model comparisons, are publicly available on the Open Science Framework (https://osf.io/2n9s8/).

## Results and discussions

**Reproduced interval.**   To ascertain the accuracy of the reproduced interval of older and younger adults, we analyzed the reproduced interval between the sixth and seventh beats for all trials. We only analyzed these intervals since the act of synchronizing to a beat and reproducing it is thought to involve different processes; one can be altered in one but not the other [50]. While the cue to tap obtained from the external stimuli during synchronization involves different error-correcting mechanisms [51] as well as the reaction time to tap after the presentation of an external stimuli, only the internal time-keeper controls and aids the motor aspect of the tapping during continuation. This dissociation between synchronization and continuation is supported by neural differences as well [52]. One older participant was eliminated due to their misunderstanding of the task.

Fig 2 illustrates the behavioral performance of both age groups where the interval was presented in the auditory or visual modality (Fig 2A). We normalized reproduced time of each interval (Fig 2B) by dividing the reproduced duration by its actual duration. These results were analyzed using a Bayesian repeated-measures ANOVA. The best performing model against the null model was found to be that of "Modality + Timing + Age + Modality × Age + Timing × Age" ($BF_{10}$ = 60.759, $error$ = 5.202).

As expected, reproduced intervals differed based on the modality in which the interval to be reproduced was presented in as well as its interval length. Our results proving differences in Modality ($BF_{Inclusion}$ = 1.877e$^{11}$; $F(1,27)$ = 40.516, $p < .001$) illustrates that reproduced durations were much more accurate in the auditory domain than when presented in the visual modality. These match previous studies which found that auditory dominance persists in temporal cognition, while visual modality tends to be under-reproduced [17, 18, 53, 54]. Differences in modality is also in line with the fact that auditory and visual system do not engage the same timing mechanisms when synchronizing to an external beat [42].

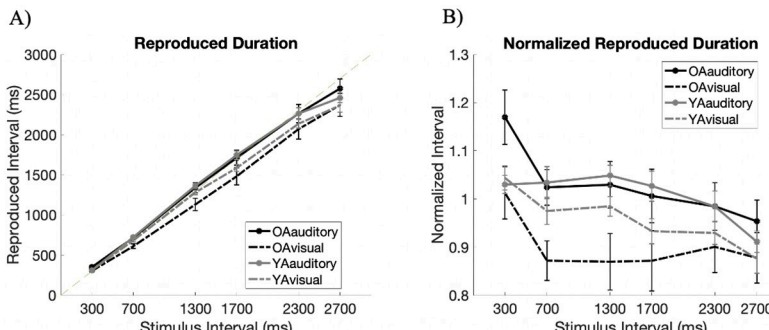

**Fig 2. Experiment 1—Reproduced intervals of older and younger adults.** These are shown in black and gray, respectively. The solid lines illustrate the reproduced intervals of those marked by auditory stimuli, and the dotted lines illustrate the reproduced intervals of those marked by visual stimuli. A) The reproduced rhythm of the intervals marked by auditory and visual stimuli for both older and younger adult groups. B) Normalized reproduced intervals that were calculated by dividing the reproduced durations by the actual interval to be reproduced. Values below one indicates the under-reproduction of the intervals. Error bars denote standard error.

Moreover, as it is to expect, both older and younger adults exhibited increases in reproduced duration when the marked temporal intervals were longer, but as the intervals got longer, these intervals were incrementally under-reproduced ($BF_{Inclusion}$ = 8.401e$^{10}$; $F(5,135)$ = 10.721, $p < .001$). Not only were longer intervals expected to be under-reproduced due to the recruitment of less automatic processes [15] but also, these can be due to other confounding factors, such as the impatience or difficulty to delay a response, and wanting to finish the experiment faster [7]. Similar trends have also been observed in foreperiod tasks where the subject's ability to predict time deteriorates with increase in interval length [45, 55]. Thus, physical properties of the stimulus can influence the way in the intervals are reproduced.

As for the effect of age, no clear evidence proving age-related differences in the reproduced interval was found. Though the best performing model includes this factor, the analysis of effects anecdotally supported the null hypothesis ($BF_{Inclusion}$ = .406; $F(1, 27)$ = .144, $p = .707$). This Bayes Factor points to no age-related differences for the general functioning of the clock, which is consistent with the general notion that the decline in performance of some temporal tasks is spared in aging [7–10].

Despite no age-related differences in general, there were age-related differences in specific to the physical stimulus properties. First, there was strong evidence for the interaction of "Modality × Age" ($BF_{Inclusion}$ = 90.851; $F(1, 27)$ = 8.036, $p = .009$). Age-related differences were magnified when the interval was presented in the visual modality (Fig 2B). While visual modality was greatly under-reproduced in both older and younger adults, it was done so in a larger extent in older adults.

Furthermore, older and younger participants differentially reproduced intervals of varying lengths. There was moderate evidence for the interaction of "Timing × Age" ($BF_{Inclusion}$ = 5.431; $F(5,135)$ = 2.195, $p < .058$). That is, the amount of under-reproduction in younger adults incremented as interval lengths got longer while the under-estimation did not necessarily increase with longer intervals in older adults. Though older adults are shown to be incapable of maintaining preparation over long temporal delays [40] and may have difficulty maintaining temporal intervals in memory [5, 8], the gradual under reproduction especially in younger adults may also be indicative of other factors such as impatience to finish the task.

Thus, the analysis conducted on the normalized reproduced intervals showed that while older adults may preserve general accuracy in reproducing rhythmic intervals, when considering the physical stimulus properties, there are age-related differences. These age-related differences depends on the modality and interval lengths and hence, it may suggest different aging effects for the distinct temporal processing centers distributed throughout the brain, as it is assumed in the distributed timing model [11, 12].

**Coefficient of variance (CV).** Given that the age-related differences in the accuracy of the reproduced durations differed with the stimulus properties, consistent with the distributed timing model [11, 12], we investigated whether the age-related difference in the coefficient of variance (CV), measure depicting scalar property [1, 11], would also be dependent on the stimulus properties. Therefore, we also calculated the CV for both age group to investigate whether the level of internal noise was higher in older adults compared to the younger cohort depending on the physical stimulus properties. The CV for each participant was calculated by dividing the standard deviation of the reproduced intervals by the actual mean.

Consistent with previous studies, the scalar property of temporal representation was observed. The analysis of effects revealed anecdotal evidence supporting the null hypothesis for Modality ($BF_{Inclusion}$ = .417; $F(1,27)$ = 1.616, $p = .214$) and strong evidence supporting the null hypothesis for Timing ($BF_{Inclusion}$ = .014; $F(5,135)$ = .580, $p = .715$). While the effect of modality cannot be completely excluded with certainty due to anecdotal evidence, studies such

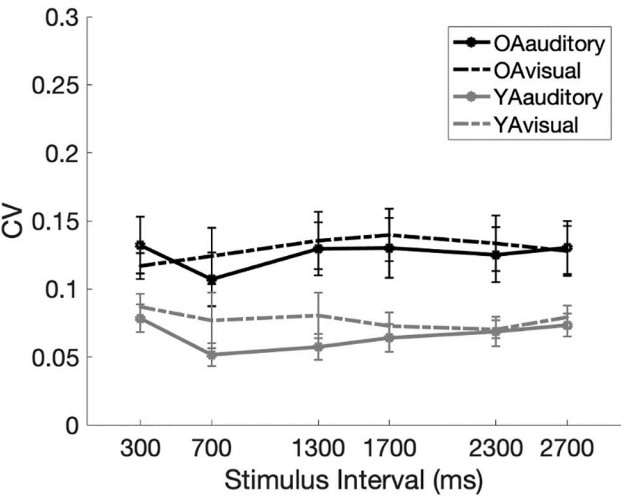

**Fig 3. Experiment 1—The coefficient of variance of two age groups given the modality and interval durations.** The solid lines illustrate the reproduced intervals of those marked by auditory stimuli, and the dotted lines illustrates the reproduced intervals of those marked by visual stimuli.

as that of Wearden and Lejeune [56] have also found that the variability of reproduced interval increases in proportion to the mean reproduced time.

Results also demonstrated that noise levels are generally higher in older adults (Fig 3). The significant main effect was observed for Age ($BF_{Inclusion}$ = 37.864; $F(1,27)$ = 14.164, $p < .001$), which, compared with the null model, is consistent with the very strong Bayes factors of the model that includes the factor of Age ($BF_{10}$ = 38.970, $error$ = 1.112%). This is consistent with previous studies [5, 7, 8, 37] that show higher variance in the performance of older adults. Hence, despite preserved accuracy, noise levels, as indicated by CV, were generally higher in older adults, even when rhythmic stimuli were used.

However, the CV was not higher in older adults at specific timings or modalities, as shown by the strong evidence for the interaction effect of "Timing × Age" ($BF_{Inclusion}$ = .033; $F(5, 27)$ = .482, $p$ = .789) and the moderate evidence for "Modality × Age" ($BF_{Inclusion}$ = .228; $F(1, 27)$ = .433, $p$ = .516), supporting the null hypothesis. Similar results were also found in Wearden and Lejeune [56] where they claim that scalar properties can be observed in older adults. In other words, while the CV was generally higher for older adults, it did not differ with temporal intervals nor modality.

Thus, results in Experiment 1 suggests that while general accuracy is relatively preserved in older adults, there are age-related differences depending on some properties of the stimulus. However, while older adults had increased variability compared to young adults, this was not dependent on the modality nor the length of intervals in which the rhythmic stimulus was presented in.

## Experiment 2

To investigate whether aging differentially affects the temporal processing of stimuli presented with different duration lengths and modalities, Experiment 1 used a simple rhythmic reproduction task. This simple task was deemed appropriate because memory capacity (eg. reproducing a previously presented standard single interval) and additional attentional resources (apart from those recruited by the physical properties of the stimulus) is not tested within the task.

Though we utilized rhythmic sequences to eliminate stress on working memory and attentional resources recruited by the task itself [36, 57], we also aimed to explore whether differences with age could expand by increasing the need for attentional resources and overloading memory capacity. Thus, in Experiment 2, we utilized the same rhythmic stimuli but also added a secondary task to increase task difficulty. As the attentional gate model explains, a dual task impairs reproduced intervals such that durations are underestimated when attending to another task [27, 58]. Nevertheless, it is unclear whether a dual task can differentially impact performance in rhythmic tasks in older and younger adults, and whether these may differ with stimulus properties. Therefore, in the following experiment, we maintained the rhythmic sequences, but added a memory task to increase the complexity of the task and explore whether these tasks would magnify age-related differences in rhythm reproduction.

## Methods

**Participants.** Fifteen older adults (*M* = 73.6; *SD* = 3.33; female = 8, male = 7), eight of whom participated in Experiment 1 a few months earlier, and 15 newly recruited younger adults (*M* = 22.2; *SD* = 2.14, female = 4, male = 11) participated in this experiment. The older adults were recruited from the Third Generation Human Resource Center in Meguro-ward (Tokyo, Japan), and the younger adult participants were recruited from The University of Tokyo. All 30 participants reported normal auditory sensitivity, and all older adults scored over 27 (*M* = 29.3; *SD* = .96) on the Mini-Mental State Examination (MMSE).

All subjects provided written informed consent, in accordance with the Declaration of Helsinki. The protocol was approved by the institutional review board of The University of Tokyo, and the subjects were given monetary awards for their participation.

**Stimuli presentation.** Stimuli of the previous reproduction task (Exp. 1) were adopted in this experiment with some minor modifications. First, to reduce the entire duration of the task, three auditory or visual stimuli, rather than five, marked the sequence of the regular beat. Second, we manipulated the frequencies of the auditory beeps and the sizes of the visual circles to include a secondary working memory task. We created five different sets of auditory and visual stimuli. Auditory beeps could either be presented at 3000Hz, 4000Hz, 5000Hz, 6000Hz, or 7000Hz with a sampling rate that is triple to that of the frequency, while visual circles were presented at diameters of either 5.45˚, 7.25˚, 9.07˚, 10.87˚, or 12.67˚ with a viewing distance of 50cm. We considered these increases in values to be sufficient for perceptual noticeability.

**Procedure.** Participants took part in four different experiments of the reproduction task, each lasting approximately 15 minutes: (1) a single reproduction task marked by auditory stimuli, (2) a single reproduction task marked by visual stimuli, (3) a dual reproduction task marked by auditory stimuli with an additional working memory task, and (4) a dual reproduction task marked by visual stimuli with an additional working memory task.

As in the previous reproduction task, participants continued the rhythm of the regular sequence of stimuli (Fig 4) and pressed the "Enter" button on the number pad at the timing of the absent fourth and fifth stimuli. This reproduced interval was then analyzed.

In addition to the reproduction task, participants also compared the stimulus characteristics between trials. After participants pressed the "Enter" button to mark the rhythm, a specific screen appeared, asking participants to compare the stimulus characteristics. For the auditory condition, participants responded to whether the beeps they heard in the present trial were higher, lower, or equivalent in pitch to those of the previous trial. Similarly, for the visual condition, participants responded to whether the circles they saw in the current trial were bigger, smaller, or equivalent in size to those of the previous trial. These trials appeared sequentially

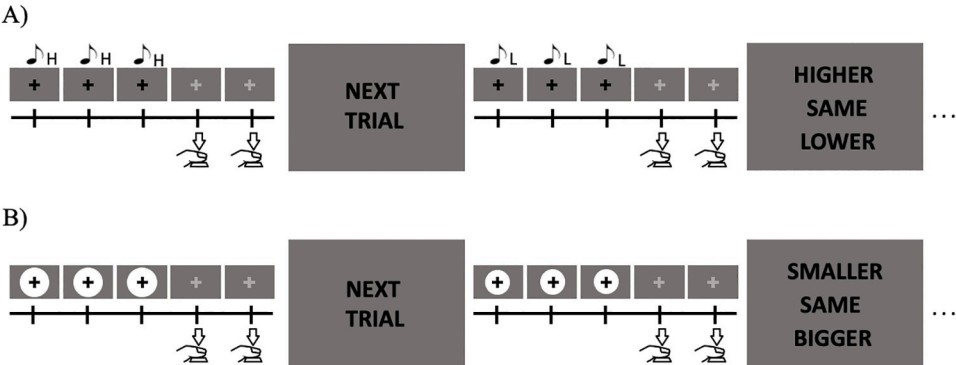

**Fig 4. Experiment 2—The procedure of the reproduction task with that of the comparison task as the secondary task.** Participants continued with the presented rhythm and remembered its stimulus characteristics: A) Auditory modality: compare the pitch (H- high, L-low); B) Visual modality: compare the circle sizes. These stimuli appeared continuously in a way that the third trial (not illustrated in the figure) required participants to compare stimulus properties with the second trial, fourth trial compared with the third trial, and so on. Note that for the presentation of the rhythm in the visual modality, the white disks flashed on and off three times. In addition, the fixation cross was maintained statically on the screen for both modalities. The instructions were displayed in Japanese in the actual experiment.

until the end of the block such that the third trial had to be compared with the second, fourth trial with the third, and so on.

Participants completed six of these blocks so that 60 trials for each experiment were performed. Each block contained ten trials, and these blocks were separated into short (under 1 sec: 300ms, 700ms), average (between 1 and 2 secs: 1300ms, 1700ms), and long durations (over 2 secs: 2300ms, 2700ms), as in the previous reproduction task.

Moreover, we semi-counterbalanced the experiment order for each participant: eight participants of each group started with the auditory task, and seven participants started with the visual task. Within the auditory and visual task, single and dual tasks were counterbalanced as well, such that eight participants began with the single task and seven participants began with the dual task. To avoid fatigue effects, subjects were freely permitted to take breaks as necessary between blocks and between experiments. After both tasks of one modality were completed, participants rated the difficulty of the tasks from a scale of one (easy) to seven (hard) with considerations of all previous experiments.

## Results and discussions

**Working memory task.** To confirm whether participants were equally engaged in the working memory task, performance for both working memory tasks (auditory and visual) for older and younger adult groups were calculated as a percentage of correct answers. The descriptive results are shown in Table 1. We excluded the data for the auditory and visual tasks

**Table 1. Average difficulty rating and standard deviation of each age group for both the single and dual tasks, in which one represents easy and seven represents hard.** Scores represent the average performance and standard deviation of the group in the working memory task.

| | | Auditory | | Visual | |
|---|---|---|---|---|---|
| | | Rating | Score | Rating | Score |
| Older (N = 14) | Single | 3.14 (SD 1.099) | | 3.85 (SD 1.460) | |
| | Dual | 4.43 (SD 1.651) | 85.05% (SD 0.096) | 4.76 (SD 1.369) | 83.60% (SD 0.100) |
| Younger (N = 15) | Single | 2.73 (SD 1.223) | | 3.13 (SD 1.187) | |
| | Dual | 4.47 (SD 1.356) | 94.71% (SD 0.064) | 5.33 (SD 0.976) | 86.42% (SD 0.056) |

of one older participant due to below chance-level performance in the memory task of the auditory modality (48.1%). We also verified that all participants rated the high-working memory task to be more difficult than the low-working memory task.

Moreover, we conducted a Bayesian Repeated Measures ANOVA on JASP [46] on the rating scores with Age as a between subjects factor, and Memory Load (Single vs Dual) and Modality (Auditory vs. Visual) as a repeated measures factor. Model comparison result shows that the best performing model was that of "Age + Memory Load + Modality + Memory Load × Age" ($BF_{10}$ = 8.443e$^{13}$, *error* = 3.021%).

In specific, analysis of effects in Bayesian statistics as well as its frequentist form, suggested that the rating scores were higher in the dual compared to single task ($BF_{Inclusion}$ = 6.064e$^{12}$; $F(1, 27)$ = 7.867, $p$ = .009) and higher in the visual compared to auditory task ($BF_{Inclusion}$ = 104.646; $F(1, 27)$ = 124.628, $p < .001$) with very strong evidence. Although there was no clear evidence for the main effect of Age ($BF_{Inclusion}$ = .424; $F(1, 27)$ = .113, $p$ = .739), there was moderate evidence for the interaction effect of "Memory Load × Age" ($BF_{Inclusion}$ = 7.550; $F(1, 27)$ = .055, $p$ = .817). The Bayes Factor suggests a tendency that younger adults rated the dual task to be harder than the single task to a greater magnitude than the older adults. Although this was unexpected, it could be that the younger adults were more aware of the function of the dual task or older adults had the tendency to say it was easier than it was.

**Reproduced intervals.** Similar to the previous experiment, we conducted a Bayesian repeated measures ANOVA as well as its frequentist equivalent on JASP [46] using the normalized reproduced interval between fourth and fifth beat. To clarify, we also included the first trial of every block in our analysis since even in this situation, participants had to actively encode and memorize the stimulus characteristics. To test whether aging affects the temporal processing of stimuli that is expressed in visual modality and supra-second intervals more than auditory and sub-second intervals in different memory loads, we used Age as a between subject factor, and Modality, Timing, and Memory Load as a repeated measures factor.

The analysis of effects revealed that the main effect of Memory Load was anecdotal ($BF_{Inclusion}$ = 1.698; $F(1, 27)$ = 3.925, $p$ = .058) in favor of the alternative hypothesis. In other words, it is unclear whether the performance worsened with increased task difficulty (Fig 5). However, given that subjective ratings were also higher in the dual task compared to the single task in the working memory task, we consider memory load to have had some sort of effect

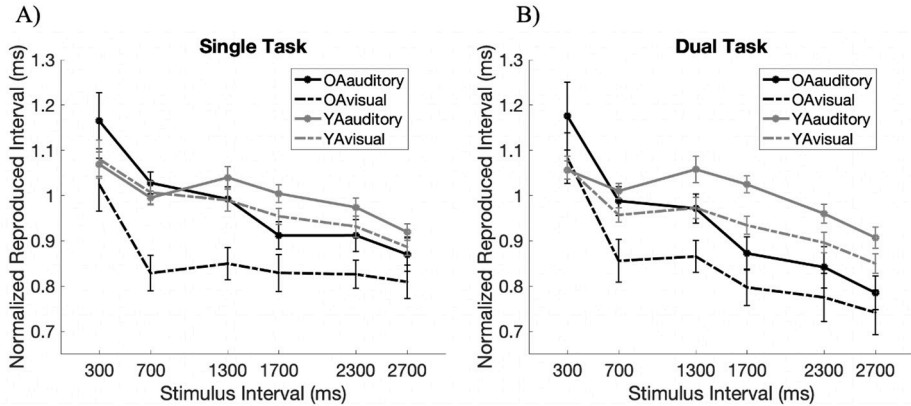

**Fig 5. Experiment 2—Normalized reproduced intervals of older and younger adults when it was A) the single task, and B) the dual task.** The solid lines illustrate the reproduced intervals of those marked by auditory stimuli, and the dotted lines illustrate the reproduced intervals of those marked by visual stimuli. Values below one represent the under-reproduction of the interval in the dual task relative to the single task. Error bars denote standard error.

despite anecdotal evidence. This is also in line with the model comparison results in terms of the Bayesian statistics. The outperforming model, compared to the null model, was "Memory Load + Modality + Timing + Age + Modality × Age + Timing × Age" ($BF_{10}$ = 1.511e$^{61}$, error = 2.889%) which indeed adds Memory Load as one of the factors that can improve model performance. In fact, when comparing this model to the second-best performing model "Modality + Timing + Age + Modality × Age + Timing × Age"; $BF_{10}$ = 9.513e$^{60}$, error = 10.433%), it can be said that there is a 59% increase in the model performance when the factor of Memory Load is added into the model. As for the interaction between "Memory Load × Age" ($BF_{Inclusion}$ = .130; $F(1,27)$ = .065, $p$ = .801), however, moderate evidence suggests no age-related differences in the effect of memory load.

Irrespective of whether task difficulty worsened task performance or not, the main effect of Modality ($BF_{Inclusion}$ = 2.320e$^{12}$; $F(1, 27)$ = 40.636, $p <$ .001) and Timing ($BF_{Inclusion}$ = 5.587e$^{46}$; $F(5, 135)$ = 24.827, $p <$ .001) were supported by strong evidence. This suggests that durations are greatly under-reproduced especially in intervals presented in the visual modality and those with longer durations.

In addition, we were interested to see whether increasing task difficulty would magnify age-related differences in the reproduction of the intervals. Unlike the reproduction task without added memory load, results revealed moderate evidence that supports the main effect of Age ($BF_{Inclusion}$ = 7.210; $F(1, 27)$ = 9.591, $p$ = .005) which suggests that older adults tend to under-reproduce more than younger adults. As for the interaction of Age with other factors, we hypothesized that age-related differences would increase when reproducing the longer intervals and for those marked by the visual modality. This was proven to be true for both Bayes Factor revealed strong evidence for the interaction effect of "Timing × Age" ($BF_{Inclusion}$ = 77655.404; $F(5, 135)$ = 3.272, $p$ = .008) and "Modality × Age" ($BF_{Inclusion}$ = 43.828; $F(1, 27)$ = 7.282, $p$ = .012). Thus, it can be said that age-related differences are indeed magnified with increased cognitive load that is present in the longer intervals and visual stimuli.

**Coefficient of variance.** Given the results of Experiment 1 where scalar property was maintained between modality and durations, we hypothesized that dual tasks would not increase the level of noise depending on stimulus properties. Yet, based on Wearden and Lejeune's findings stating that task difficulty violates scalar properties [56], we also hypothesized that the additional task could decrease the precision in the reproduced interval overall due to memory overload and a limitation in the attentional resources allocated to the timing task.

The effect of the dual task on the reproduced interval for both modalities in the multiple durations was explored using Bayesian repeated measures ANOVA as well as its frequentist form on JASP [46]. Results from Bayesian model comparisons revealed that the best performing model describing the behavioral results (Fig 6) was that of "Modality + Age + Modality × Age" ($BF_{10}$ = 2.609e$^{6}$, error = 4.373) and the analysis of effects for the inclusion of the factor of Memory Load was anecdotal ($BF_{Inclusion}$ = 0.418; $F(1, 27)$ = 2.409, $p$ = .132).

In contrast to our hypothesis, however, variability depended on the stimulus properties. Analysis of effects on Modality ($BF_{Inclusion}$ = 6729.911; $F(1, 27)$ = 15.156, $p <$ .001) suggests that the coefficient of variance was higher for the visual modality. This means that unlike in the simple rhythmic task in Experiment 1, in Experiment 2, precision was reduced when the stimulus was presented in the visual modality. Illustratively, these results show that while an increased attentional and memory load did not significantly increase the variability, precision was lower for the visual modality.

As for the effects of aging on the variability of the reproduced intervals, results showed, in accordance with our previous experiment, that older adults are generally more variable than younger adults ($BF_{Inclusion}$ = 460.707; $F(1, 27)$ = 24.465, $p <$ .001). Furthermore, in contrast to

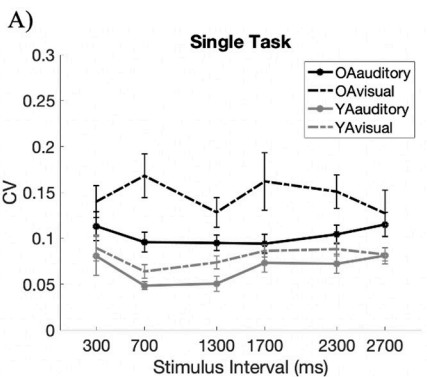 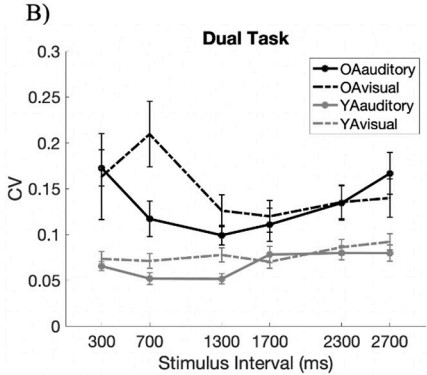

**Fig 6. Experiment 2—The effect of dual task on the coefficient of variance of older and younger adults in A) the single task, and B) the dual task.** Modalities are illustrated using the solid lines for auditory and the dotted lines for visual. Error bars denote standard error.

the reproduction task without memory load, the best performing model also includes the interaction effect of "Modality × Age". Yet, we cannot make strong conclusions on whether older adults are more variable in visual modality compared to younger adults for the analysis of effects only reveals anecdotal evidence supporting this interaction ($BF_{Inclusion}$ = 1.213; $F(1, 27)$ = 3.011, $p$ = .094) despite the strong evidence for the main effect of Modality ($BF_{Inclusion}$ = 6729.911; $F(1, 27)$ = 15.156, $p < .001$). In fact, adding the interaction of "Modality × Age" only improves model performance by 12%. Thus, while results indicate that older adults are generally more variable in their reproduced duration, it is unclear whether they suffer more than younger adults when the intervals are presented in the visual modality.

## General discussions

In this study, we considered aging to affect behavioral performance in timing tasks through the decline in cognitive factors. To delineate whether this was true, we investigated age-related differences in pure temporal cognition by minimizing attentional and memory components using rhythmic sequences. We conducted two types of experiments which demanded attentional resources in different aspects of the tasks. First, in Experiment 1, we delimited the attentional resources to those solely placed on the stimulus properties and removed any attentional resources demanded by the task by using rhythmic sequences. We considered both the differences in modality and interval length under the distributed timing model, to investigate whether it is possible that rhythmic auditory, or sub-second intervals are preserved in older adults while that of visual or longer intervals are not. Second, in Experiment 2, we investigated the effects of the dual task to determine whether increasing the cognitive demand in the task magnifies age-related differences in temporal reproduction in particular to different stimulus properties.

### Reproduction accuracy and aging

Under the centralized model, it has been said that older adults generally have clocks that pace at a slower [2, 3, 59], faster [5–7] or equal rate [8–10] as those of younger adults.

The performance of older adults in our rhythm reproduction task, along with our previous work on implicit timing [60], showed that aging does not dull temporal prediction and production except when additional cognitive loads are added to the task.

In our study, participants were required to predict temporal intervals using information retrieved from the fixed rhythmic interval presented prior to the cue to continue tapping. Previous studies with foreperiod tasks have shown that older adults have impaired performance in tasks that require temporal preparations for uncertain events, but do not have deficits in fixed intervals nor do they take more time when preparing their response [39, 40]. Given that our rhythmic task required the intact ability of temporal preparation in older adults, we asked participants to reproduce intervals presented in the auditory and visual modality using rhythmic sequences. We eliminated any additional cognitive resources required in different timing tasks apart from those of the physical stimulus properties.

Investigating modality differences in the accuracy of rhythmic timing in older adults was key to this study due to the growing research about modality differences in the mechanisms behind sensorimotor synchronization [42] and the distributed timing model [11–13]. Our study was different from those of Lustig and Meck [20], and McAuley and colleagues [26], who also investigated modality differences in temporal processing between older and younger participants, since they used single intervals. Not only do single intervals require different brain structures compared to those of rhythmic timing for its processing [61], but it also requires participants to maintain temporal interval duration in their working memory in order to reproduce it or judge them. In the case that older adults have deficits in cognitive abilities [29, 62], they are at a clear disadvantage in this timing task and are more likely to under-reproduce the intervals. Thus, in our study we controlled the extra cognitive load from the single-interval reproduction task by using rhythmic sequences. We isolated the temporal cognition from other higher cognitive factors and found that there were no age-related differences in the general accuracy in which the intervals were reproduced.

As in Turgeon and Wing [2] and Bangert and Balota [5], we used rhythmic sequences to reduce the cognitive load from the tasks. While Turgeon and Wing [2] found overall over-reproduction of intervals in older adults, they had their participants internally generate what they believed a specific interval would be. Since the internal generation of a beat increases activation in the basal ganglia [35], which declines with age [63], it seems more likely that age-related differences in performance using this approach can be observed. Moreover, the study of Bangert and Balota [5] differs in that participants synchronized and tapped to a particular beat for three minutes. This methodology, on the other hand, allows differences in age to be magnified not only due to increased demands in sustained attention, but also the effect of fatigue on temporal preparation [64]. Therefore, we deemed important to add to these studies by asking older adults to simply continue tapping to the rhythmic beat for a short duration. Our task minimized the requirements for additional resources for the maintenance of the interval in working memory for more than one cycle, or the use of sustained attention to continue reproducing for longer periods.

Though we do not claim that we reduced all working memory and attentional resources from the timing task, when we minimized the working memory and attentional load to the best of our ability, there was not enough evidence supporting age-related differences in temporal prediction and processing (Experiment 1). The lack of strong evidence suggests that there may be many individual differences. Nevertheless, when we added a secondary task to the simple rhythmic task, age-related differences appeared, though there was no interaction between aging and the task difficulty (Experiment 2). The age-related differences found in could be explained by the attentional gate model in which the limited attentional resources are placed on the working memory task rather than that of the temporal task, leading to differences in the number of pulses sent to the accumulator [27]. In other words, while aging may seem to have some effect on temporal cognition in specific to modalities and temporal intervals, it may be the overloaded capacity of the working memory to maintain the interval, and the restricted

amount of attentional load imposed on the timing task, that cause differences in the clock speed that otherwise pulses at a rate comparable to that of young adults.

## Effects of interval length with age

Irrespective of the task, we considered that interval length of a rhythmic stimulus was a stimulus property that took up more working memory and required increased attentional load. Thus, we hypothesized that the cognitive load required in rhythmic beats with longer inter-stimulus intervals would magnify age-related differences in accuracy. Our results proved this to be true, even with rhythmic tasks.

Our hypothesis was based on the idea that short and long durations recruit different mechanisms [11, 14, 15] and that, unlike the short intervals that tend to be processed by sensory mechanisms, longer intervals recruit attention and other cognitive functions [15, 21]. Moreover, it has been shown that even in simple reaction time tasks, older participants are not able to maintain temporal preparation over long delays [40]. Thus, it is plausible that aged individuals reproduce even shorter intervals in dual tasks compared to younger adults, especially with longer time intervals. Such behavioral performance could also be explained by the deterioration of cognitive abilities due to structural changes in the brain for older adults [29]. Moreover, it is also very likely that rhythmic intervals of long duration suffers more from cognitive load especially because for sequences of sounds to be perceived as rhythmic, they should have an interval that is not too long [65].

Our results indicated that there are age-related differences when asked to reproduce longer intervals. When we performed this study, we considered the lack of age-related differences because Turgeon and colleagues [37] stated that older adults can perform tasks as well as younger adults if they can recruit other unimpaired neural networks. Similar observations has been made recently in Droit-Volet and colleagues' study [66] where older adults adopted the inclusion of hazard function when processing temporal information, leading to better performance than that of younger adults in the timing task. In fact, in our study, the increase in the amount of under-reproduction in older adults as intervals get longer does not seem to be as significant as that of younger adults, especially in the simple rhythmic reproduction task. When the cognitive load was added, however, the amount of under-reproduction in older adults seems to have increased as intervals became longer. Our guess is that this may be due to the impatience to finish the task, especially in younger adults in the simple reproduction task, but also for older adults in the dual task, so that they can answer to the working memory task quickly.

## Differential effects in modality with age

Another stimulus property we considered was that of modality. Our results showed that differences between modalities may be more substantial in the older adult group compared to the younger group. In other words, in conjunction with the distributed timing model, there seems to be age-related changes in the clocks that are dedicated to specific modalities.

In our study, we found that reproduced durations differed with modality for both older and younger adults. Under the distributed timing model, differences in temporal processing of visual and auditory temporal stimuli are explained by alterations in the clock speed of the different modalities. Compared to auditory signals, visual signals tend to be more slowly initiated with counting of the pulses that the pacemaker emits [54]; thus, visual signals are reproduced for shorter amounts than auditory signals [17, 18, 53]. Moreover, these differences can also be due to modality specificity that stems from sensory cortices [22, 67] with auditory dominance in rhythmic beats relative to the visual modality [23]. As such, it is a widely accepted fact that perceived and reproduced durations differ with modality.

Nevertheless, our study contributes to a growing body of literature on the dissociation between modalities. Even when additional cognitive load of the task was removed, older adults seemed to under-reproduce more when the stimulus was presented in the visual modality. This was also true when older adults under-reproduced the visual modality even more than younger adults despite the rating of the task being similar to or even rated a bit easier than the younger adults.

The tendency to judge visual stimuli as even shorter than auditory stimuli in a secondary memory task [20] is known already as well as the fact that aging renders adults to be vulnerable to divided attention in visual tasks [26]. Yet, our study adds that this is true not only in the case of judging and reproducing single intervals [20, 26], but also in rhythmic sequences which, as argued previously, recruit less attentional resources than single intervals, as well as different regions of the brain [68]. The fact that these modality differences remained even when using rhythmic sequences suggests that aging affects production of intervals.

### Aging and temporal precision

Furthermore, our study shows that age-related differences were observed in the variability of the reproduced intervals. It has been widely accepted that with increasing age, people are more variable [5, 7, 8, 37].

Though older adults were more variable, the level of noise did not increase with specific stimulus properties. First, precision was not related to the length of the temporal interval despite an increased level of controlled attention necessary for the processing of longer intervals. Similar to the findings of Piras and Coull [16], which indicate similar variance in the millisecond and second range in predictive timing, we showed that older adults maintained the same scalar property at the level of milliseconds and seconds when tapping at an isochronous rhythm.

Second, we considered the possibility that the stimulus property of modality could alter the level of precision in the rhythmic task. Previous studies have shown that auditory and visual modalities are shown to have differential levels of noise. For example, Wearden and Lejeune [56] demonstrated that the scalar property of auditory and visual modalities do not superimpose. This was also true for Zélanti and Droit-Volet [69], which noted that children of young age exhibited reduced sensitivity to visual stimuli relative to auditory stimuli. However, in our study, simply using visual modality to present the rhythmic stimuli did not manifest in higher coefficient of variance, even in older adults. Yet, when the difficulty of the task was increased by adding a secondary working memory task, this modality difference appeared in both older and younger adults, and probably to a greater extent in older adults.

The fact that there are modality differences between the variability of the reproduced durations is suggestive of the fact that increased variability in older adults may not be simply due to the motor components of the tasks since these should be equal irrespective of modality. In fact, the attentional load and working memory capacity demanded by the task, rather than the stimulus property itself, could be a factor that increase this variability.

### Limitations

In this study, we did not use standardized psychometric instruments such as that of Dundee stress state questionnaire (DSSQ) to assess the motivation and stress before and after the task despite these factors contributing to performance.

Nevertheless, we took two main precautions to maximize performance. First, duration of a single task was a total of 15 minutes and participants could take breaks during the task as necessary. Older adults in specific were limited to only two tasks per day, with breaks between tasks

as well. After each task, all participants were asked to come out of the experiment booth to talk about their opinion and comments about the task. There seemed to be nothing about the task that indicated fatigue or worry, such that many participants experienced the task to be "fun, as a game". Second, we limited the tapping duration to reproducing one interval unlike that of Bangert and Balota [5] which lasted three minutes. We did not record succeeding intervals mainly to avoid fatigue, reduce the load of focused attention, and memory decay of the interval.

Because psychological measures may have been important given that even the simplest timing task may be mentally exhausting [64] and that the older population may not be as accustomed to the experimental testing as the university students, future study could investigate whether these psychological measures can explain the age-related differences observed. Despite this, we also believe that psychological measures of stress, motivation and worry would not impact our result by much given participants comments and rating scores.

## Conclusions

In conclusion, these results question the idea that older adults have a generally slower internal clock than younger adults. General accuracy of their reproduced interval suggests that the speed of their pulses is comparable to those of younger adults when they are asked to do a rhythmic task. Accuracy, however, is shown to be altered in older adults by the increased attentional component of the stimuli modality and rhythmic interval length. We also confirm that older adults are more variable when reproducing intervals, and that these do not depend on the stimulus properties. When task difficulty was increased with a secondary task, however, older adults were less accurate and modality differences seems to have appeared in precision as well. These findings indicate that (1) even in an inherently automatic rhythmic task, where older and younger adult's general accuracy is comparable, accuracy but not precision is altered by the stimulus properties, and (2) an increase in task load can magnify age-related differences in both accuracy and precision.

## Acknowledgments

We acknowledge Meguro Ward Silver Human Resource Center for the recruitment of older participants.

## Author Contributions

**Conceptualization:** Yuko Yotsumoto.

**Data curation:** Elisa M. Gallego Hiroyasu.

**Formal analysis:** Elisa M. Gallego Hiroyasu.

**Funding acquisition:** Yuko Yotsumoto.

**Investigation:** Elisa M. Gallego Hiroyasu.

**Methodology:** Elisa M. Gallego Hiroyasu, Yuko Yotsumoto.

**Project administration:** Yuko Yotsumoto.

**Resources:** Yuko Yotsumoto.

**Supervision:** Yuko Yotsumoto.

**Validation:** Elisa M. Gallego Hiroyasu, Yuko Yotsumoto.

**Visualization:** Elisa M. Gallego Hiroyasu.

Writing – **original draft:** Elisa M. Gallego Hiroyasu.

Writing – **review & editing:** Elisa M. Gallego Hiroyasu, Yuko Yotsumoto.

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
