## [Decision Letter · Decision Letter 0]

18 Jan 2021

PONE-D-20-38134

Disentangling the effects of modality, interval length and task difficulty on the accuracy and precision of older adults in a rhythmic reproduction task.

PLOS ONE

Dear Dr. Yotsumoto,

Thank you for submitting your manuscript to PLOS ONE. After careful consideration, we feel that it has merit but does not fully meet PLOS ONE’s publication criteria as it currently stands. Therefore, we invite you to submit a revised version of the manuscript that addresses the points raised during the review process.

We look forward to receiving your revised manuscript.

Kind regards,

Michael B. Steinborn, PhD

Academic Editor

PLOS ONE

Editorial Comment: I could find one referee to comment on your work, so I decided to take the effort of reviewing myself. Detailed comments and suggestions can be found below. Reviewer 1 raised some methodological issues that would prevent publication in its present form. To name some issues, the statistics are not transparently presented, the experimental design is somewhat underpowered, and theoretical mechanisms should be given more emphasis. My own comments are provided below and are aimed to aid you in the process of revision.

(-1-) Power/Sample Size

(--) I agree with Reviewer 1 that the present design is not sufficiently powered. Very strictly, this would mean that further data collection would be necessary (though I would not demand it). However, I suggest finding ways handling the issue of power in the revised version of the manuscript.   

(--) The reporting of statistics might be reworked in the revised version of the manuscript. At present, it is not transparently presented. I personally suggest using classic frequency-based statistical analyses to verify results (though I would not strictly demand it). 

(-2-) Self-report measures

Given that age-related differences in performance measures are determined by both cognitive ability and motivation (task engagement), it is essential (at least common) to collect both pre-test and post-test measures of subjective state. This is not mentioned in the present study. With this respect, I would like to suggest a psychometric instrument to assessing stress state in performance settings, which has been regarded the gold standard in many research domains. The dundee stress state questionnaire (DSSQ, Langner et al., 2010, for methodical aspects of assessing time-related processing effects on engagement/distress) is a theory-oriented instrument aimed to assess the fundamental dimensions of subjective state in performance settings, namely task engagement, distress, and worry, which can further be divided into more specific sub-facets. The measure is widely accepted and the instrument is well-evaluated, and has good psychometric properties. I would recommend the DSSQ for future studies, but more importantly, it would be appreciable if the authors could give a short opinion or outlook on the possibilities of assessing engagement to the task in future studies and to elaborate somewhat more deeply on potential limitations with this regard.

(-4-) Theoretical mechanisms

I would also agree with Reviewer 1 that the theoretical background needs to be reworked during the revision of the manuscript. To this end, I have prepared a list of relevant papers that might be important to consult during the revision process and to consider in the revised version of the manuscript. 

(--) relatively new methodological work on developmental effects of time-related processing

Vallesi, A., McIntosh, A. R., & Stuss, D. T. (2009). Temporal preparation in aging: A functional MRI study.  Neuropsychologia, 47(13), 2876-2881. doi:10.1016/j.neuropsychologia.2009.06.013

Mento, G., & Granziol, U. (2020). The Developing Predictive Brain: How Implicit Temporal Expectancy Induced by Local  and Global Prediction Shapes Action Preparation Across Development. Developmental Science.  doi:10.1111/desc.12954

(--) frequently cited literature on methodical aspects of temporal processing 

Bherer, L., & Belleville, S. (2004). Age-related differences in response preparation: The role of time uncertainty. Journals  of Gerontology Series B-Psychological Sciences and Social Sciences, 59(2), P66-P74.

Langner, R. et al. (2010). Mental fatigue and temporal preparation in simple reaction-time performance. Acta Psychologica, 133(1), 64-72. doi:10.1016/j.actpsy.2009.10.001

Steinborn, M. B., Rolke, B., Bratzke, D., & Ulrich, R. (2008). Sequential effects within a short foreperiod context: Evidence  for the conditioning account of temporal preparation. Acta Psychologica, 129(2), 297-307.  doi:10.1016/j.actpsy.2008.08.005

Journal Requirements:

Reviewers' comments:

Reviewer's Responses to Questions

**Comments to the Author**

1. Is the manuscript technically sound, and do the data support the conclusions?

Reviewer #1: Partly

2. Has the statistical analysis been performed appropriately and rigorously? 

Reviewer #1: I Don't Know

3. Have the authors made all data underlying the findings in their manuscript fully available?

Reviewer #1: Yes

4. Is the manuscript presented in an intelligible fashion and written in standard English?

Reviewer #1: Yes

5. Review Comments to the Author

Reviewer #1: I carefully read this paper by Yotsumoto and colleagues, which investigates how aging impairs temporal processing capacity. In this regard, the authors have adopted an experimental task of rhythmic temporal reproduction through different sensory modalities and also manipulating the cognitive load. The topic is potentially interesting even if difficult to generalize to applicative aspects or simply to everyday life. The introduction is clear, linear and generally well written although I have noticed some typos and errors in English. However, I fear that there are some major concerns that currently limit the ability to accept this article for publication.

1) A first aspect concerns the statistical approach. I admit that I am not a Bayesian analysis expert, which is why my judgment only reflects subjective perplexities. In general, I appreciate the idea of using alternative methods to simple GLM models. However, I wonder if the use of a Bayesian approach is adequate for a study that presents several experiments with even a complex factorial design (and very low sample size, see below). For example, in experiment 2 the authors state that

" The analysis of effects revealed that the main effect of Memory Load was anecdotal (BF = 1.698). In other words, it is unclear whether the performance worsened with increased task difficulty (Fig 5). However, the outperforming model, compared to the null model, was that of " Memory Load + Modality + Timing + Age + Modality × Age + 61 Timing × Age " (BF10 = 1.511e, error = 2.889%) which indeed adds Memory Load as one of the factors that can improve model performance. In fact, when comparing this model to the second-best performing model “Modality + Timing + Age + Modality × Age 60 + Timing × Age "; BF10 = 9.513e, error = 10.433%), it can be said that there is a 59% increase in the model performance. Thus though analysis of effects revealed anecdotal”.

Personally, I find it difficult to understand the validity of this statement since it is not clear to me in which way a more complex model can substitute a null result regarding a fundamental manipulation for the logic of the study, that is, the absence of effect of the double task. I therefore wonder if it is not the case to add an additional analysis using classical or mixed models (GLMM) in order to better frame the results. As I am not an expert on the matter, however, I refer the decision to the editor.

2) Another important issue is the very low sample size. I realize that the study has already been recorded, so an accurate analysis of the statistical power has already been done, as suggested by the G Power test done by the authors. However, I cannot help but find it contradictory to accept a study with so few subjects (14 per experiment and per group) since one of the strategies strongly proposed by the OSF is to increase the sample size to improve data replicability. Among other things, this point is not trivial especially in the presence of experiments or groups characterized by a high individual variability, as in the case of the elderly group. This issue is even more important if considering that it deals with simple behavioral experiments rather than invasive or complex neuroimaging techniques or special populations like patients or very young infants.

3) It is not clear to me why the authors considered only a single interval between the sixth and seventh stimuli rather than, for example, an average of all intervals. I think that a data analysis taking into account this aspect could be more robust.

4) Another potentially relevant aspect is the type of task used to test the distributed model. The perceptual discrimination task actually loads the working memory more than it does for attention, since the difficulty of this task is low. Therefore, I wonder if the results may actually reflect a "loss" of beats as predicted by the model and hypothesized by the authors or rather memory overload. Even if the task used by itself does not impact much on the memory, nevertheless the reproduction of rhythms involves in any case a coding maintenance recovery of the temporal information from the memory. Therefore, I am not sure that the manipulation proposed in this study is actually suitable for testing the distributed model, especially since it involves elderly people, who could also have difficulties in the mechanisms of coding and retrieving information in memory rather than simply attention load.

Minor issues:

Pag 16, 331: “I thik that it is Exp. 2”

6. PLOS authors have the option to publish the peer review history of their article (what does this mean?). If published, this will include your full peer review and any attached files.

Reviewer #1: No

---

## [Author Response · Author response to Decision Letter 0]

22 Feb 2021

please see "Response to Reviewers".

---

## [Editor Report · Decision Letter 1]

24 Feb 2021

Disentangling the effects of modality, interval length and task difficulty on the accuracy and precision of older adults in a rhythmic reproduction task.

PONE-D-20-38134R1

Dear Dr. Yotsumoto,

We’re pleased to inform you that your manuscript has been judged scientifically suitable for publication and will be formally accepted for publication once it meets all outstanding technical requirements.

Kind regards,

Michael B. Steinborn, PhD

Academic Editor

PLOS ONE

Additional Editor Comments:

I read the manusript again and commented on some occasions, however, this will likely not take much time to consider. I think, it is not necessary to invite further rounds as these points are only very minor and can be handled during the final manuscript preparation stage.

 (--) check for typos, examples, --p. 2, line 28-30, numbering in brackets (1)--p.9,  line 167, statistical measures in italics (e.g., M, SD, p, F, etc., applies here and on other occasions) (--) p. 34, line 724, provide more explanation, briefly, suggestion: this is to ensure that the observed age-related differences are not biased by differences in mental states immediately before the testing. For example, age groups often are recruited from a normal population not experienced with experimental testing, while younger groups are mostly based on student samples having high experience with testing. .. (--) references--check references for typos--doi number is lacking on some occasions
---

## [Editor Report · Acceptance letter]

8 Mar 2021

PONE-D-20-38134R1 

Disentangling the effects of modality, interval length and task difficulty on the accuracy and precision of older adults in a rhythmic reproduction task. 

Dear Dr. Yotsumoto:

I'm pleased to inform you that your manuscript has been deemed suitable for publication in PLOS ONE. Congratulations! Your manuscript is now with our production department. 

Kind regards, 

on behalf of

Dr. Michael B. Steinborn 

Academic Editor

PLOS ONE